# Identification of Villages' Development Types Using a Comprehensive Natural–Socioeconomic Framework

**Yaqiu Liu [1], Jian Liu [1], Can Guo [1], Tingting Zhang [2], Ailing Wang [1],* and Xinyang Yu [1],***

[1] College of Resources and Environment, Shandong Agricultural University, Taian 271018, China; 2018010084@sdau.edu.cn (Y.L.); liujian200810@163.com (J.L.); CanGuo2021@163.com (C.G.)

[2] Shandong Tiancheng Land Planning and Design Institute Co., Ltd., Jinan 250014, China; suztting@126.com

* Correspondence: ailingwang@sdau.edu.cn (A.W.); yuxinyang19860915@163.com (X.Y.)

**Abstract:** The establishment of a comprehensive framework to identify village development types is crucial to formulate plans for rural development and promote rural revitalization. This study proposed a natural–socioeconomic framework to identify the types of villages based on field survey, statistical data, and multi-source remote sensing images. The framework was constructed by combining the two-dimensional natural suitability/restriction evaluation and the four-dimensional socioeconomic development level evaluation. Then, the modified multiplication-weighted summation method and the coupling coordination degree algorithm were employed to identify the villages' development types. A total of 774 villages of the Laiyang County, eastern China were used as the study areas to examine the framework. The results demonstrated the following. (1) There were 243,318 and 151 villages with high, moderate, low natural suitability, and 62 villages with natural restrictions; and 158,366 and 250 villages with high, moderate, and low economic development level, respectively. The distribution characteristic of natural evaluation was "high in the southwest and low in the northeast", and the socioeconomic development level was generally centered on the urban area, which presented a "high–medium–low" circle-layer distribution structure. (2) There were 247 villages with high-level coupling coordination, 464 villages with intermediate coupling coordination, 1 village with low-level coupling coordination, and 62 villages with disordered coupling. (3) Based on the coupling coordination evaluation results, villages in the study area were grouped into five types: urbanization development (31%), construction development (16%), agglomeration linkage development (27%), decrease and improvement development (18%), and relocation and integration development (8%). The framework of villages' development types identification established in this study can enrich the theory of rural geography, and the applied research results can provide a basis for rural revitalization and development planning.

**Keywords:** comprehensive natural–socioeconomic framework; multi-source data; natural suitability/restriction; socioeconomic development level; coupling and coordination

## 1. Introduction

The rural village is a natural and social complex formed over time that couples the ecological environment and socioeconomic systems [1]. Sustainable development is determined by a set of socioeconomic and natural environmental factors for rural areas. Different countries and regions have different policies for rural development due to different levels of development. In order to improve the rural living environment and the living standard of rural residents as well as reduce the gap between urban and rural development, many countries developed sustainable development strategies [2] and programs using different methodological approaches.

In China, rural villages are experiencing prominent population loss and socioeconomic recession, especially those with harsh natural environments and remote locations [3]. Imbalanced development between urban and rural areas and inadequate rural development have become two major societal concerns. The Chinese government puts forward

the strategy of rural revitalization and proposes that rural village development should be promoted according to the development status, location conditions, and resource endowments of different villages in accordance with the law and evolutionary trends of village development [4].

Therefore, from the comprehensive perspectives of the natural suitability and the level of socioeconomic development, it has become an important issue for China to implement the strategy of Rural Revitalization. Taking Laiyang County of Shandong Province as an example, based on field investigation, statistical data, and multi-source remote sensing images, this paper combines two-dimensional natural suitability/restriction with a four-dimensional socioeconomic development level, constructs a comprehensive natural–socioeconomic framework, identifies and analyzes the development types of villages, and puts forward constructive development suggestions, Thus, it enriches the theoretical framework and system of village classification development, and it also provides a scientific basis for making regional village development decisions.

The structure of this paper is as follows. Section 2 provides a literature review and proposes the research direction of this study. Section 3 presents the construction of the comprehensive natural–socioeconomic framework, which forms the theoretical support for this study. Section 4 introduces the study area, data sources, research methods, and processing. Section 5 presents an analysis of the results of the natural suitability/restriction evaluation, socioeconomic development level evaluation, coupling coordination between natural suitability and socioeconomic development level, and villages' development types identification, and it also discusses the theoretical contributions and future prospects of this study. Section 6 presents the conclusions.

## 2. Literature Review

Recent studies on rural areas mainly focus on the neo-endogenous rural development [5], social innovation in rural governance [6], critical drivers for sustainable development in rural areas [7], the potential using efficiency of rural development [8], rural development and natural resource management [9], urban–rural development [10], rural sustainable development [11,12], rural revitalization [13,14], rural transformation [15], rural tourism [16–18], wisdom [19,20], and beautiful village construction [21]. The research area is distributed in different countries and regions, including Europe, Russia, and post-Soviet countries, Italy, Korea, America, and China. Different characteristics of the natural environment and regional socioeconomic conditions lead to great differences in the development of villages. Therefore, the development types of villages should be identified so as to detect the feasible development and revitalization paths to better guide the direction of village development. Research has been conducted on the identification of villages' development types. For instance, the conditions of village relocation and consolidation were identified by constructing the suitability evaluation indexes system from three aspects (landscape index, topographical conditions, land utilization situation) [22]. In addition, some studies focused on the evaluation of villages' development levels [23], e.g., the village development evaluation indexes system was constructed from three aspects of production, life, and ecological function; or from four dimensions of living resources, industrial base, regional transportation, and cultural ecological environment, to detect the types of villages [24,25].

Most of the rural economic indicators involved in the evaluation indexes system are based on the conventional economic statistical indicators, such as the per capita disposable income of residents [26], economic density, total agricultural output value [27], per capita total output value of agriculture [28], ratio of non-agricultural output value [29], and development level of village collective economy [30]. Since the 1990s, the nighttime light remote sensing data were gradually used to objectively reflect the level of social and economic development of a region [31]. Especially, the Luojia1-01 image can reflect social and economic development in finer spatial resolution, which is one of the most professional types of nighttime light remote sensing data [32], and it was used to detect the development of constructed areas, e.g., county-level regions in Hubei, Hunan, and Jiangxi [33]. However,

the application of Luojia1-01 is temporally concentrated in urban areas [34], involving population spatialization [35], population distribution [36], economic index prediction [37], etc. The application of nighttime light remote sensing images in rural areas remains to be an emergent task. Therefore, it is indispensable to explore the application of Luojia1-01 nighttime light images in rural socioeconomic development evaluation.

Studies on rural development employed the suitability or development levels of a single aspect to identify the villages' development types. However, the village is the coupling of natural environment and social economy; a clear and systematic framework combining natural and socioeconomic factors is indispensable to the identification of the villages' development types. According to the coupling and coordinating relationship between the natural environment and social economy, we should build a comprehensive framework and divide the development types of rural villages. The coupling coordination degree model is a model originating from physics, which is used to evaluate the coupling coordination relationship between different systems. It is currently a reference for measurements by different systems or system coupling coordination between internal elements. However, the existing studies were merely limited to new urbanization, rural settlement layout [38–40], etc., which is difficult to be extended to the identification of villages' development types. Hence, it is important to strengthen the direction of study that couples the suitability and development of village development types. Furthermore, it is the uppermost priority of the current research to explore the implementation path suitable for China's rural development.

Thus, this study strived to proposed a comprehensive natural–socioeconomic framework of villages' development types with natural–socioeconomic factors and multi-source data. The study was conducted and examined in the villages of Laiyang County, eastern China. Comprehensive consideration of natural suitability and socioeconomic development, the modified multiplication-weighted summation method, and the coupling coordination degree model were used to establish the comprehensive natural–socioeconomic framework, which was used to identify the villages' development types scientifically and then put forward rural revitalization suggestions. The study was expected to provide a reference for promoting the rural revitalization and village planning.

## 3. Comprehensive Natural–Socioeconomic Framework

The formation and development of rural villages are influenced by the natural environment and socioeconomic systems. Rural development and revitalization should be based on the natural environmental conditions of villages, which provide basic conditions for the development of villages. Meanwhile, such villages also have more populations, higher incomes, and better infrastructure. If the natural conditions are not adequate for development (excessive slopes in the terrain or being prone to geological disasters), the villages will be restricted and will gradually decline or even die out. Therefore, there is usually a certain coupling between the natural environmental conditions of villages and their socioeconomic development. In addition, due to the differences in transportation, economic input, and management, rural socioeconomic development levels are different. The conventional economic development factors are not sufficient to reflect the level of economic development. As the nighttime light images can reflect the regional economic level objectively, they should be included to construct the framework to identify the type of rural village development.

The comprehensive natural–socioeconomic framework (Figure 1) was constructed as the following steps. Firstly, on account of the natural environment conditions, factors such as slope, altitude, water resources, geological hazards, and ecological functional areas were selected; the modified multiplication-weighted summation model was used to evaluate the suitability and restriction of rural development. When a natural factor was worse than a certain value and unsuitable for human habitation, it was determined as a restricted development area. In other regions, the suitability level of village development was evaluated according to the levels of natural factors affecting human productivity and

life. Secondly, from the perspective of socioeconomics, the high-grade road, historical and cultural resources, total population, collective income, average light density at night, education, and medical treatment were selected from four dimensions of traffic location, resource endowment, cluster scale, and basic facilities, to evaluate the socioeconomic development of villages. Finally, the coupling coordination degree model was employed to evaluate the coupling coordination degree of rural natural suitability and socioeconomic development. Based on the evaluation results of coupling coordination degree, natural suitability, and socioeconomic development, the village development types were assigned, and then differentiated suggestions for rural development and revitalization were put forward.

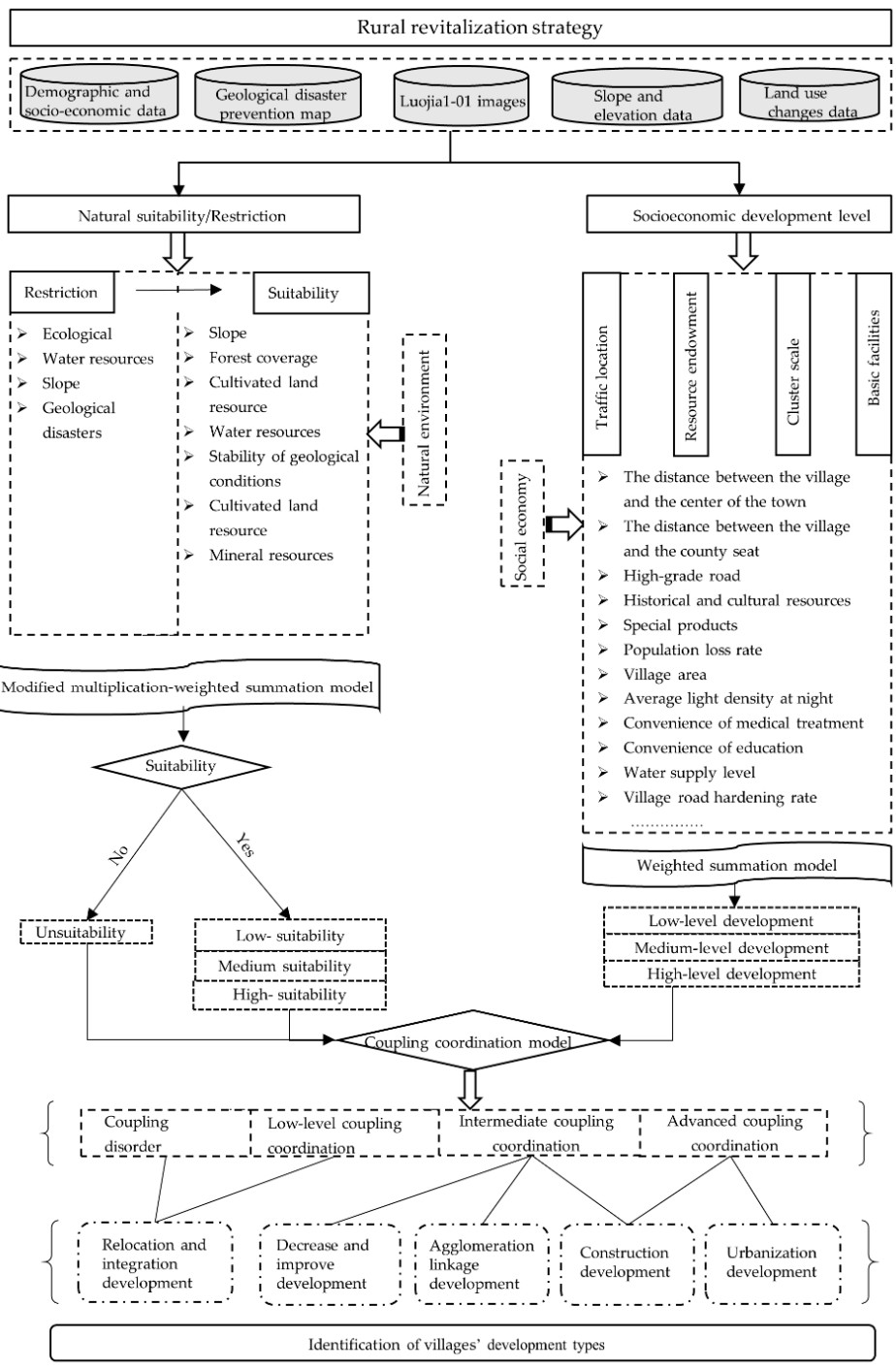

**Figure 1.** Comprehensive natural–socioeconomic framework.

## 4. Methodology and Data Source

### 4.1. Study Area

Laiyang County (120°31′06″−120°59′12″ E, 36°34′25″−37°09′46″ N) is located in the east of Shandong province, eastern China (Figure 2). The study area contains 18 towns (784 villages) with a permanent population of 872,000 and an area of 1732 km², in which the cultivated land area is 830 km². It is a low hilly area with a maximum elevation of 375 m. The terrain slopes from north to south, among which 68.7% is low-mountainous hilly and 31.3% is flat. It has a temperate monsoon sub humid climate, adequate light, and four distinct seasons. The territory is rich in mineral resources, mainly minerals, among which the reserves of bentonite and zeolite rank second in the province. It is known as the hometown of Chinese pears, Chinese dinosaurs, calligraphy, and peanut oil. Since ten villages are urban villages, they were not within the scope of this study. Therefore, 774 villages were finally identified as the research objects.

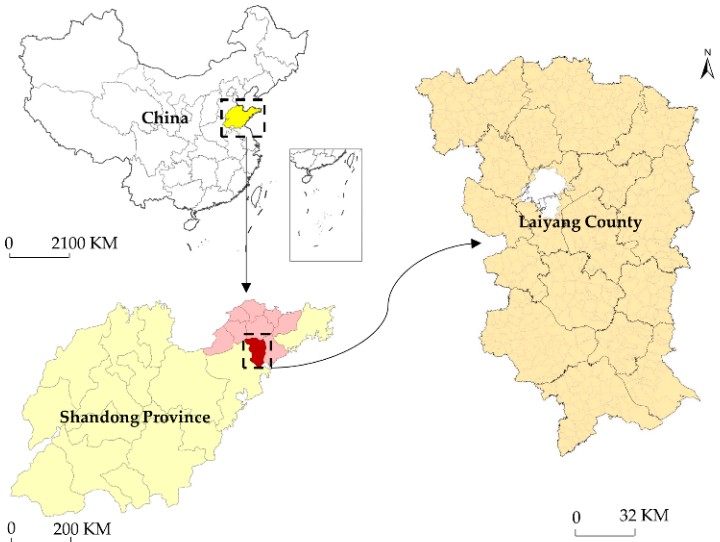

**Figure 2.** Location of the study area.

### 4.2. Data Source

The data sources include: (1) Laojia1-01 nighttime light images, which were from Hubei data and an application network of a high-resolution earth observation system and processed by Envi5.3 and ArcGIS 10.7; (2) remote sensing images of Laiyang County's geographical national conditions survey 5–6 in 2019, land-use change data in 2018 and a geological disaster prevention map of Laiyang County, which were obtained from the Laiyang Natural Resources and Planning Bureau; (3) demographic and socioeconomic data, which were from the Laiyang Bureau of Statistics, the Statistical Yearbook of Laiyang County, Social, and Economic Statistical Bulletin of Laiyang County and field questionnaire; (4) slope and elevation data were mainly derived from the geospatial data cloud of the Chinese Academy of Sciences.

### 4.3. Evaluation of Natural Suitability/Restriction

#### 4.3.1. Selection of Evaluation Indexes

Villages are generally developed and formed in regions with flat terrain, rich resources, good geology, and ecological security. Within a certain range, the smaller the slope and elevation, the more convenient are production and life, and thus, it is more conducive to the formation and development of the village. The forest land coverage rate reflects the ecological quality, and the higher the forest land coverage rate, the better the ecological environment quality. Natural resources such as water, arable land, and minerals form the material basis of rural development. The richer the natural resources, the greater the potential for rural development. Good geological conditions and low risks for disasters

are conducive to long-term rural development. However, villages are not suited to areas with bad natural conditions, such as excessive slopes, excessive elevation, proneness to geological disasters, being located in ecological protection areas, being too close to water, and being prone to floods. Therefore, these villages' development should be restricted.

Laiyang County is located on low-mountain and hilly land with a peak altitude of 375 m. It is rich in arable land, water, and mineral resources, but their distribution is uneven. Considering the natural suitability, restrictions, and natural environment conditions, seven evaluation indexes, i.e., the slope, altitude, geological conditions, forest coverage, cultivated land resources, distance from river/reservoirs and mineral resources were selected to evaluate the natural suitability/restriction of villages, and five indexes (important ecological areas, distance from water, altitude [41], slope, and risk of geological disaster) [42] were selected to evaluate restrictions for village development (Table 1).

**Table 1.** Indexes and weights of natural suitability/restriction evaluation in the study area.

| Category | Indicators | Weight | Description | Attribute [1] |
|---|---|---|---|---|
| Restrictive | Ecological area | - | (whether it was ecological important functional area) Yes = 0, No = 1 | - |
| | Water resources | - | (whether the distance from the river reservoir was less than 500 m) Yes = 0, No = 1 | - |
| | Slope | - | (whether the slope was greater than 25°) Yes = 0, No = 1 | - |
| | Geological disasters | - | (whether geological disasters were highly prone to occurrence) Yes = 0, No = 1 | - |
| Suitability | Slope | 0.25 | $<2° = 4, 2°–6° = 3, 6°–15° = 2, 15°–25° = 1$ | + |
| | Elevation | 0.25 | Classification by elevation value | − |
| | Forest coverage | 0.08 | Forestland coverage rate = forestland area/total village area | + |
| | Cultivated land resource | 0.08 | Ratio of cultivated land = total area of cultivated land in village | + |
| | Water resources | 0.13 | The distance from the river reservoir was 500–1000 m = 4, 1000–1500 m = 3, 1500–2000 m = 2, > 2000 m = 1 | + |
| | Geological conditions | 0.13 | No risk = 3, low risk = 2, medium risk = 1 | + |
| | Mineral resources | 0.08 | Potassium feldspar and iron ore = 4, soil ore = 3, rock brick and stone = 2, none = 1 | + |

[1] "+" A positive index, indicating that the higher the score, the better the index; "−" A negative index, indicating that the lower the score, the better the index.

### 4.3.2. Quantification of Evaluation Indexes

According to the requirements of village development based on natural environment conditions, the restrictions and suitability factors were quantified by referring to relevant standards and literature [43]. First, the restrictive factor was quantified, and the score of the restrictive factor was set to 0 or 1. If a limiting factor reached an extreme value and restricted the village's construction, it was assigned 0, and the area was directly designated as an unsuitable area. An area below the extremum value was set as 1; it then continued to participate in the suitability evaluation. According to the influences of suitability factors on village construction and rural development, considering the reality of Laiyang County, the suitability factors were graded and assigned, and the extreme value method was adopted to conduct standardization treatment. The relative importance of each suitability factor on village construction and rural development was compared, and the weight of each suitability index was determined by analytic hierarchy process (AHP). A total of 15 experts, including professors and experts in land use, urban and rural planning land planning, land assessment and resource remote sensing, as well as representatives of natural resources bureau, planning institute, and other relevant units, were invited to score the indicators. Based on the scoring of all experts, the judgment matrix is constructed to make the weight

quantitative on the basis of qualitative data, and the result is more objective. The indexes and weights for natural suitability evaluation are shown in Table 1.

### 4.3.3. Construction of Modified Multiplication-Weighted Summation Evaluation Model

This study took the village as the unit and constructed a modified multiplication-weighted summation evaluation model to evaluate suitability/restriction. It includes the following two parts:

Firstly, we used the series multiplication model [44] to evaluate restriction. As long as the index value of one factor exceeded the limit value, a village was classified as an unsuitable development zone and a restricted construction zone. The specific formula is shown in Equation (1), which was shown below:

$$S_x = \prod_{k=1}^{m} A_{kx} \tag{1}$$

where $S_x$ is the restriction index of a village $x$; $m$ is the number of limiting value evaluation factors; $k$ is the order of limiting value factor; and $A_{kx}$ is the score of the $k^{th}$ limit value factor of a village $x$.

For the other villages, the natural suitability index was calculated by the weighted summation model [45] to evaluate their natural suitability. The calculation formula is shown in Equation (2), as follows:

$$S_y = \sum_{i=1}^{j} B_{iy} \times b_i \tag{2}$$

where $S_y$ is the suitability index of a village $y$, $j$ is the number of suitability evaluation factors; $i$ is the order of suitability factor; $B_{iy}$ is the score of the $i^{th}$ suitability factor of a village $y$; and $b_i$ is the weight of the $i^{th}$ suitability factor.

### 4.4. Evaluation of Socioeconomic Development

#### 4.4.1. Selection of Evaluation Indexes

The socioeconomic development level of villages is mainly reflected in traffic location, resource endowment, cluster scale, and basic facilities. Good traffic location facilitates the flow of people and resources. The higher the grade of road, the easier it is to travel, and the more conducive to the development of a village. The closer a village is to a town, the easier it is to be driven by its development. The endowment of social resources, such as historical cultural resources and specialty products, is formed by long-term development, which is not only a sign of the level of rural development but also an advantage for future development. The scale of village agglomeration, including the population, income level, and village size, is a sign of the level of rural development. The larger the population, the higher the income, the brighter the nightlight, and the larger the village scale, the higher the level of village development. Under the current trend of rural population outflow, excessive population outflow affects the development of villages. Village infrastructure construction includes educational conditions, medical conditions, hardened roads, water supply, gas supply, etc., which are all reflections of rural development and the quality of production and life. The more perfect the infrastructure, the more comfortable people's lives will be.

The study area has high-speed railway and various levels of highways. The travel conditions and the radiation effect of cities and towns also differ among different villages with different road grades and distances to cities and towns [46]. Laiyang pear, Wu Long calligraphy, and other cultural resources and specialty products are renowned at home and abroad; part of the villages was a historical and cultural village, dinosaur fossil village, and other specialty villages. Village cluster scale, such as population [47], economy, and land agglomeration [48], is different. The study area attaches great importance to rural

construction and has relatively good infrastructure [49], but there are differences among villages. Hence, 14 indicators were selected from four dimensions, including traffic location, resource endowment, cluster scale, and basic facilities, to construct the evaluation indexes system of rural socioeconomic development in the study area, as shown in Table 2.

**Table 2.** Weights of socioeconomic development indicators for Laiyang County.

| Criterion Layer | Weight | Indexes Layer | Weight | Calculation Method and Description [1] | Attribute [2] |
|---|---|---|---|---|---|
| Traffic location | 0.23 | The distance between the village and the center of the town | 0.17 | Calculated using the NEAR tool in ArcGIS | − |
| | | The distance between the village and the county seat | 0.29 | Calculated using the NEAR tool in ArcGIS | − |
| | | High-grade road | 0.54 | National road = 4; Provincial road = 3; County and township road = 2; Others = 1 | + |
| Resources endowment | 0.12 | Historical and cultural resources | 0.50 | (Whether it is a folk village) Yes = 1, No = 0 | + |
| | | Special products | 0.50 | (Do they have any special products?) Yes = 1, No = 0 | + |
| Cluster scale | 0.42 | Population | 0.30 | Permanent population | + |
| | | Population loss rate | 0.30 | Population loss rate = (registered residence population—resident total population)/registered residence population | − |
| | | Economic situation | 0.32 | Collective income | + |
| | | | | Average light density at night | + |
| | | Village area | 0.08 | Village area | + |
| Basic facilities | 0.23 | Convenience of medical treatment | 0.16 | Distance between the nearest hospital and the village committee | − |
| | | | | (whether there is a community health service) Yes = 1, No = 0 | + |
| | | Convenience of education | 0.16 | (whether there is kindergarten) Yes = 1, No = 0 | + |
| | | | | Distance between the nearest primary school and the village committee | − |
| | | | | The nearest junior high school and this village (residence) committee distance | − |
| | | Gas supply level | 0.08 | (Whether pipeline gas is connected) Yes = 1, No = 0 | + |
| | | Water supply level | 0.26 | Tap water = 1, bottled water = 2, purified water = 3, mineral water = 4, water purifier = 5 | − |
| | | Village road hardening rate | 0.34 | Village road hardening rate = road hardening length/total road length | + |

[1] Economic situation, "collective income" is as important as "average light density at night"; Convenience of medical treatment, "distance between the nearest hospital and the village committee" is as important as "whether there is a community health service"; Convenience of education, "whether there is kindergarten", "distance between the nearest primary school and the village committee", and "the nearest junior high school and this village (residence) committee distance" are equally important. [2] "−" A negative indexes, indicating that the lower the score, the better the indexes; "+" A positive indexes, indicating that the higher the score, the better the indexes.

### 4.4.2. Quantification of Evaluation Indicator

Considering the impacts of evaluation indicators on rural socioeconomic development, combined with the reality of the study area, the indexes values were calculated or assigned directly, and the extreme value method was adopted to standardize the indicators. Distance indexes include the quantification of the distance from a given village to the nearest town center and to the county seat. They were calculated using the near tool in ArcGIS. AHP was used to determine the weights of indicators. The quantification, weight, and attributes of each indicator are shown in Table 2.

### 4.4.3. Construction of a Weighted Summation Evaluation Model

The weighted summation model [50] was constructed to calculate the socioeconomic development indexes of villages. The formula is shown in Equation (3).

$$F_y = \sum_{i=1}^{m} \left( J_{ny} \times j_n \right) \tag{3}$$

where $F_y$ is the socioeconomic development index of village $y$; $m$ is the socioeconomic index number; $J_{ny}$ is the $n$th standardized socioeconomic indicator value of village $y$; and $j_n$ is the $n$th socioeconomic index weight.

### *4.5. Identification of Villages' Development Types*

#### 4.5.1. Construction of the Coupling Degree Model

Coupling refers to the phenomenon in which two or more systems or forms influence each other and even unite through interaction. A coupling degree model is an effective model to measure the synergistic relationship between two or more elements [51]. In this study, the coupling degree model is constructed to measure the coupling degree between the two systems of natural suitability and socioeconomic development level. The calculation is as follows:

$$C = 2 \times \left[ \frac{S \times F}{(S + F)^2} \right]^{\frac{1}{2}} \tag{4}$$

where $C$ is the coupling degree between natural suitability and socioeconomic development, $C \in [0, 1]$; S and F represent the natural suitability indexes and socioeconomic development level indexes, respectively. By referring to the existing classification standards [52] and combining them with the actual situation of the coupling degree in the study area, the coordination degree is divided into four levels: low-level, medium-level, superior-level, and high-level coupling. The classification standards are shown in Table 3.

**Table 3.** Coupling degree and coupling coordination degree division standard.

| Coupling Degree $C$ | Coordination Level | Coupling Coordination Degree $D$ | Coordination Level |
|---|---|---|---|
| 0.00–0.30 | Low-level coupling | 0.00–0.20 | Coupling disorders |
| 0.30–0.60 | Medium-level coupling | 0.20–0.40 | Low-level coupling coordination |
| 0.60–0.90 | Superior-level coupling | 0.40–0.60 | Intermediate coupling coordination |
| 0.90–1.00 | High-level coupling | 0.60–0.80 | Advanced coupling coordination |
| - | - | 0.80–1.00 | Extreme coupled coordination |

#### 4.5.2. Construction of the Coupling Coordination Degree Model

Coupling coordination degree is used to judge the coupling and coordination status between systems and among various elements within the system. In this study, the coupling coordination degree model was constructed in order to judge whether the two systems of natural suitability and socioeconomic development level are consistent and the degree of interactive coupling between the two systems, so as to better identify the coupling

coordination relationship between them. The coordination index T and the coupling coordination degree D are calculated using Equations (5) and (6).

$$T = \mu \times S + v \times F \tag{5}$$

$$D = \sqrt{C \times T} \tag{6}$$

In Equations (5) and (6), T is the coordinated index of rural natural suitability and socioeconomic development, $\mu$ and $v$ are undetermined coefficients, and $\mu + v = 1$. This study holds that rural natural suitability and socioeconomic development are equally important—$\mu = v = 0.5$. *D* is the coupling coordination degree of natural suitability and socioeconomic development. The higher the value of *D*, the better the coupling coordination of rural natural suitability and socioeconomic development. According to related studies [53,54] and according to the value of *D*, the region can be divided into five levels, namely coupling disorder, low-level coupling coordination, intermediate coupling coordination, advanced coupling coordination, and extreme coupling coordination, by using the uniform distribution method, as shown in Table 3.

## 5. Results and Discussions

### 5.1. Natural Suitability/Restriction Evaluation

The evaluation results using natural evaluation showed that different types of natural restrictions existed in 62 villages, and the natural suitability index S was 0, indicating that they were unsuitability areas (US). These villages are located in an ecological protection zone. They are close to the stream course or reservoir, and they are prone to flooding. They had large topographic variations or high risks for geological disasters. Therefore, they were considered US.

The natural suitability indexes S of the remaining 712 villages ranged from 0.23 to 0.83, and the commonly used natural break point method was used to categorize village suitability into three levels: high-suitability areas (HS), medium-suitability areas (MS), and low-suitability areas (LS). Analysis of the results found that the natural suitability of villages varied greatly, showing a gradually decreasing spatial distribution pattern from southwest to northeast, as shown in Figure 3. Among these villages, 243 villages in Chengxiang, Guliu, etc. were HS (0.51 < S ≤ 0.83). These villages had flat terrain, low geological disaster risk, rich natural resources (such as water and arable land), high forest coverage rate, good ecological environment, and the suitability degree was high. Rural development and revitalization should prioritize these areas in the future. A total of 318 villages in Fenggezhuang, Jiangtuan towns were MS (0.40 < S ≤ 0.50). Among them, Chadaokou, Xiyuquan, and other villages have relatively flat terrain, lesser risks of geological disaster, relatively abundant natural resources, good ecological environment quality, and high suitability. A total of 151 villages under the jurisdiction of towns such as north Jianggezhuang and Beimazhuang have steep slopes, complex terrain, geological hazards, a relative lack of water resources, little cultivated land, low ecological environment quality, or lower suitability. They had LS (0.23 < S ≤ 0.39).

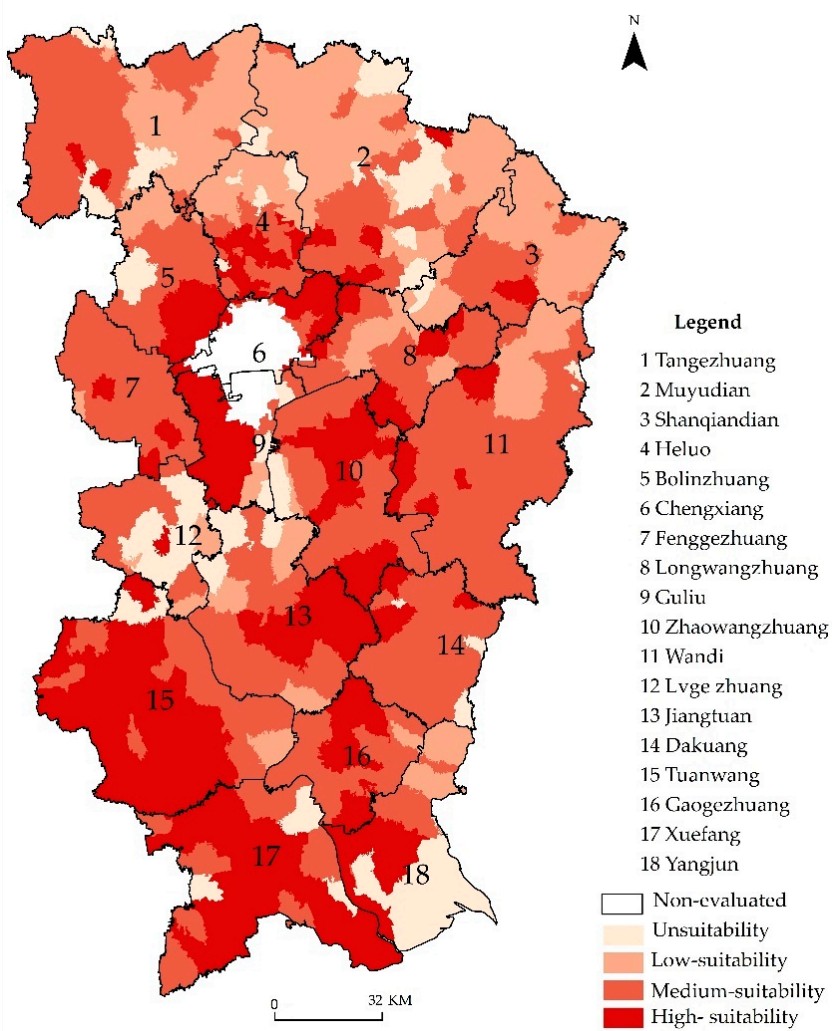

**Figure 3.** Evaluation results of village suitability in Laiyang City.

*5.2. Socioeconomic Development Level Evaluation*

The socioeconomic development index F of the villages in Laiyang County ranged from 0.12 to 0.52, and the values were grouped into three grades using the natural break point method: high-level development area (HD), medium-level development area (MD), and low-level development area (LD). It can be seen from Figure 4 that the level of rural socioeconomic development presents obvious spatial differences, and it presented a high–medium–low circle-layer distribution structure centered on the urban area. Statistical results found that there were 158 villages with HD (0.29 < F ≤ 0.52), accounting for 20% of the evaluated villages. They are mainly distributed in Guliu, Chengxiang, and Jiangtuan towns. These villages have high-level traffic conditions, large areas, proximity to the urban area, expressways, more convenient village transportation, good infrastructure, clear resource advantages, and high overall development. A total of 366 villages occupied MD (0.21 < F ≤ 0.29), accounting for 47% of the evaluated villages. They are mainly distributed in Bo Linzhuang, Heluo, Xuefang, Yangjun, and other towns. These villages had relatively good locations, relatively convenient external travel, some infrastructure, relatively abundant characteristic resources, high forest coverage rates, and relatively high development levels. Another 250 villages were LD (F ≤ 0.21), accounting for 32% of the evaluated villages. They are concentrated in the northwest of Tangezhuang; the northeast of Wandi; and the southeast of Dakuang and other towns. These villages are far away from the city, and they have large slopes, imperfect infrastructure, the advantage of average scale (population, area, etc.), incomplete public facilities, and low levels of overall development.

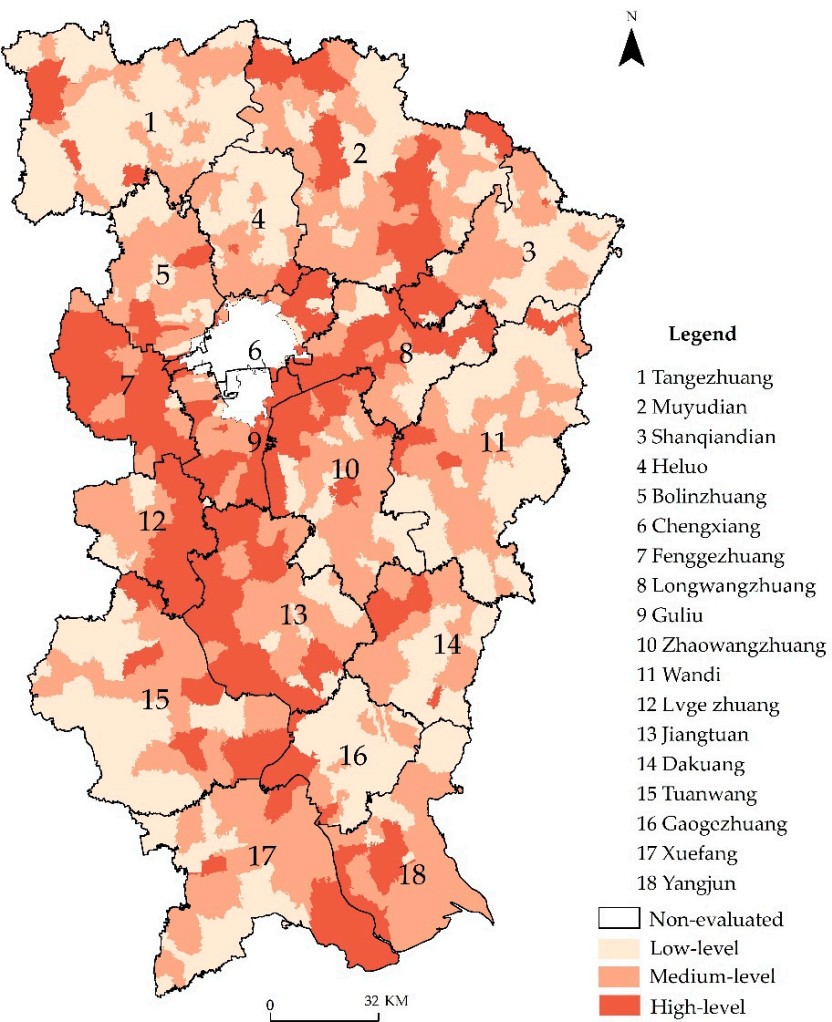

**Figure 4.** Evaluation results of villages' development in Laiyang County.

*5.3. Identification of Villages' Development Types*

5.3.1. Coupling Coordination between Natural Suitability and Socioeconomic Development Level

The average coupling degree C of the study area (Figure 5a) exceeded 0.86, indicating that the coupling degree of natural suitability and socioeconomic development level was high. According to statistics, there were 593 villages with high-level coupling, 119 villages with superior-level coupling, and 62 villages with low-level coupling. Although the amounts of suitable area in some villages in the west of Tuan Wang and Xue Fang were relatively high, the resource endowment and cluster-scale conditions were not very useful due to the large distances from the urban area, which also came with the cost of low social development levels.

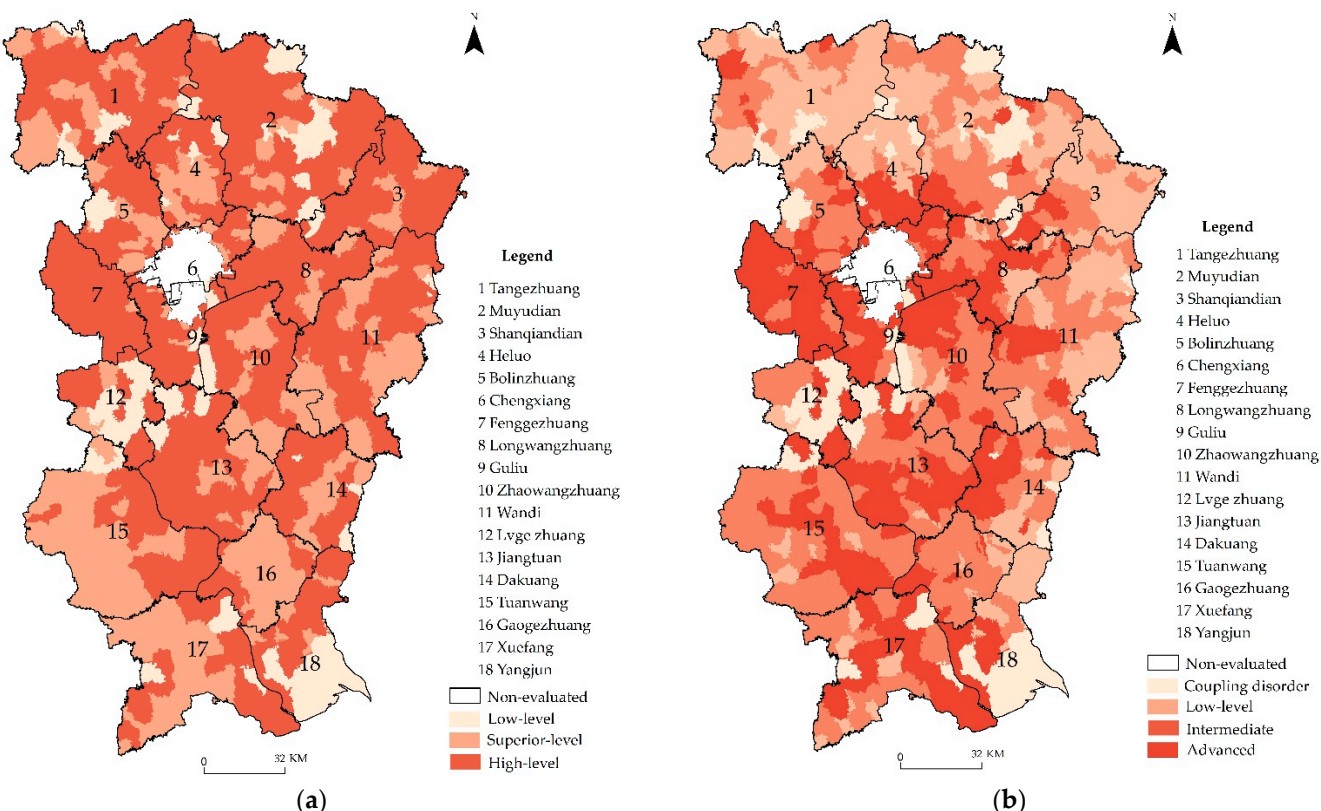

**Figure 5.** Coupling characteristics of village suitability and socioeconomic development: (**a**) Coupling degree; (**b**) Coupling coordination degree.

The coupling coordination degree D of natural suitability and socioeconomic development of the villages ranged from 0.38 to 0.71. According to division standard, there are four coupling coordination types, advanced coupling coordination of 247 villages, intermediate coupling coordination of 464 villages, low-level coupling coordination of one village, and disordered coupling of 62 villages. They had obvious spatial differences, which generally present a trend of taking the urban area as the center of the circle, gradually decreasing outwards, and high in the south and low in the north (Figure 5b). The advanced coupling coordination villages were scattered in the areas around each. Villages of the low-level coupling coordination and disordered coupling were mainly located in Muyudian, Heluo town, etc., and the suitability and socioeconomic development level of villages were low.

### 5.3.2. Villages' Development Types Identification

Integrating the four levels of natural suitability, three levels of socioeconomic development level, and four levels of coupling coordination, 17 coupling modes of natural suitability and socioeconomic development in the study area were obtained. Among them, the intermediate coupling coordination–medium suitability–medium level developmental villages were the most commonly found areas, containing 113 villages, constituting about 15% of the evaluated villages in Laiyang County. The socioeconomic development level in medium-level villages was higher than other villages, but these villages' coupling coordination degrees were higher; a change of natural suitability will drive a change of socioeconomic development level. High coupling coordination–high suitability–medium level development and intermediate coupling coordination–medium suitability–low level development villages accounted for 13% of the evaluated villages. The coupling coordination degrees in these villages were high.

Therefore, from the two aspects of rural natural suitability and socioeconomic development, combined with the degree of coupling coordination between them, the village development stages were divided into five types, using the combination matrix method

of multi-dimensional characteristics [55,56]. All types of coupling disorder (D1) and low coupling coordination (D2) villages were classified as the relocation and integration development. All villages of the intermediate coupling coordination (D3)-low-suitability were classified as the decreasing and improving development. The villages with medium and high combination in the type of advanced coupling coordination (D4) were classified as the urbanization development, and the rest of villages were classified as the construction and agglomeration linkage development based on the coupling coordination degree (Figure 6, Table 4).

The 237 urbanization development villages were located in the surrounding parts of urban areas, mainly guided by urbanization development, accounting for 31% of the evaluated villages. This type of village has obvious geographical advantages, abundant resources, a scale (population, area, etc.) advantage, regular shape, completed infrastructure, and gentle slopes. The radiation of towns and cities plays a vital role in the economic development of surrounding villages. Future development and revitalization of rural areas should rely on a better economic foundation and regional advantages, and develop green food, machinery manufacturing, health medical care, leisure tourism, logistics and other industries, so as to promote the integrated development of village industries and improve the level of ecological environment.

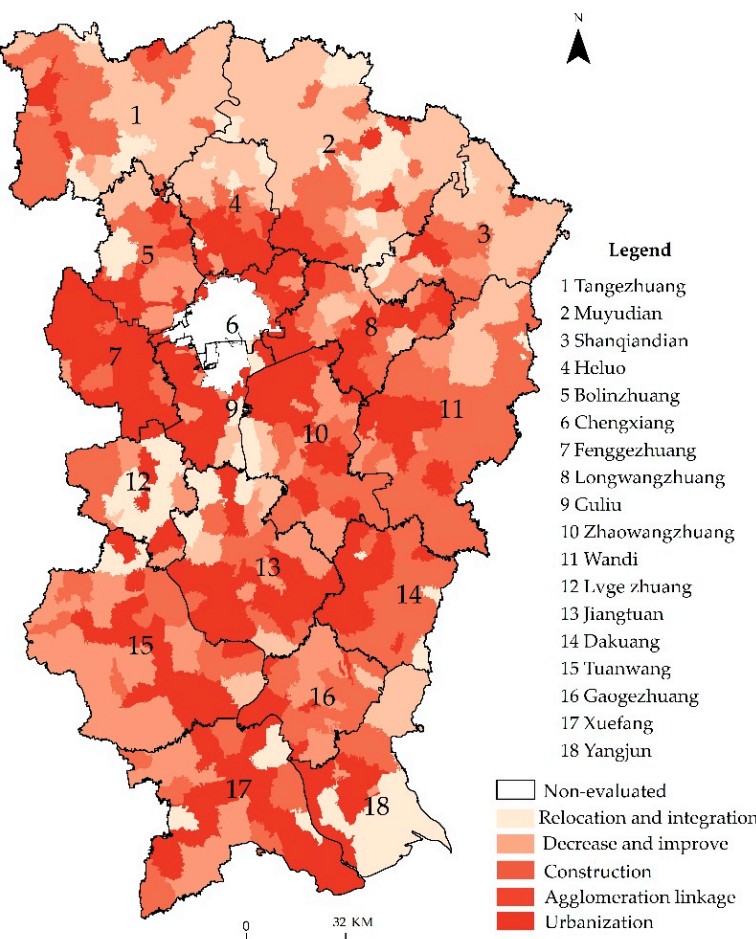

**Figure 6.** Identification of villages' development types in Laiyang County.

**Table 4.** Identification method of villages' development types in Laiyang County.

| Coupling Degree of Compatibility | Natural Suitability [1] | Socioeconomic Development Level [2] | Number of Villages | Types | Combined |
|---|---|---|---|---|---|
| Coupling disorders (D1) | US | LD | 25 | Relocation and integration development | 63 |
| | US | MD | 15 | | |
| | US | HD | 22 | | |
| Low level coupling coordination (D2) | LS | LD | 1 | | |
| Intermediate coupling coordination (D3) | LS | LD | 59 | Decrease and improve development | 140 |
| | LS | MD | 63 | | |
| | LS | HD | 18 | | |
| | MS | LD | 98 | Agglomeration linkage development | 211 |
| | MS | MD | 113 | | |
| | MS | HD | 3 | Construction development | 123 |
| | HS | LD | 67 | | |
| | HS | MD | 43 | | |
| | LS | HD | 10 | | |
| Advanced coupling coordination (D4) | MS | MD | 33 | Urbanization development | 237 |
| | MS | HD | 71 | | |
| | HS | MD | 99 | | |
| | HS | HD | 34 | | |

[1] Natural suitability, US = Unsuitable; LS = Low Suitability; MS = Medium Suitability; HS = Highly Suitable. [2] Socioeconomic development level, LD = Low-level development; MD = Medium-level development; HD = High-level development.

Construction development villages had a good economic foundation and relatively good basic facilities. The focus of development is to strengthen the construction of external transportation networks and improve the basic supporting facilities. There were 123 villages in this category, accounting for 16% of the evaluated villages. Villages of this kind are mainly distributed in Jiangtuan, Xuefang, Yangjun and other towns, with relatively abundant resources, large scale and regular shape, gentle terrain, complete basic infrastructure, and high-level development. In the future, the village development needs to optimize and perfect the external road transportation facilities, improve the external transportation convenience, and strengthen the connections with urban areas, towns and central villages. The construction of infrastructure and public service facilities should be improved to make life and production more convenient for villagers.

The natural conditions of the agglomeration linkage developing villages were moderately suitable, and the socioeconomic development was average. The key to these villages' development is to optimize the layout of the villages and develop specialty economies by relying on the advantages of unique resources. There were 211 villages in this type, accounting for 27% of the evaluated villages. This kind of village is relatively small, and it is mainly distributed in Heluo, Tuanwang, Wandi, and other jurisdictions. In this region, the village locations and traffic conditions are good, the infrastructure is relatively good, and the development level is medium; however, the relief of the terrain is large, and thus, development is limited by the natural conditions. The settlements are small and scattered, the layouts of villages are unreasonable, and the suitability is medium. In the future, such villages can merge scattered village lands through village planning so as to make villages contiguous and layout optimized. Relying on industry and trade, trade circulation, and other resources, industrial development will drive economic development and realize the combination of agriculture and industry. At the same time, some relocated villages will be attracted to move in and expand the agglomeration scale so as to drive the common development of other surrounding areas and increase the income of the collective and villagers.

The natural conditions and infrastructure of the declining and improving development villages are insufficient, and the level of economic development is different. The crux of these villages' development is to reduce the size of villages, improve village infrastructure, and develop village economies. There are 140 villages in total, accounting for 18% of the evaluated villages. Such villages are mainly distributed in the towns of Yangjun, Muyudian, and Wandi. The villages have poor natural conditions, are far from the central urban areas and towns, have low-level external traffic conditions, are small in scale, and have insufficient resource conditions and infrastructure. In order to adapt to the law of social development, this kind of village can gradually reduce the scale of village land according to the population migration and the withdrawal of homestead in the future, and use the land economically and intensively. The infrastructure and people's working and living conditions should be appropriately improved. At the same time, the government should guide rural residents to transfer their agricultural land to large agricultural households and agricultural enterprises, expand the scale of agricultural production, and develop appropriate economies of scale.

Relocation and integration development villages are not suitable for residential development, so they need to be relocated to develop primary, secondary, and tertiary industries and integrate with the new villages for development. There are 63 villages in this category, accounting for 8% of the total number of villages in Laiyang County. This kind of village is mainly distributed in towns such as Muyu Dian, Dakuang, Wandi, and Yangjun. These villages have many hidden geology disasters, low-level living conditions, distanced locations, small areas, harsh ecological environments, unsuitability for living, and continued construction of infrastructure and public service facilities. These villages should be gradually relocated and merged into nearby towns or central villages. Additionally, then, they can develop secondary and tertiary industries in order to integrate with local development. The original village land can be reclaimed for arable land, ecological land, etc., so as to develop ecological agriculture, sightseeing agriculture, high-tech agriculture, etc., and promote socioeconomic development. At the same time, the relocated villagers should be guided to transfer their farmlands to large agricultural households and agricultural enterprises, to expand the scale of agricultural production, and to develop appropriate economies of scale.

*5.4. Discussions*

The identification of villages' development types is an important part of the classification and promotion of rural revitalization, rural planning, and territorial spatial planning. Scientifically and reasonably identifying the types of village development and putting forward the corresponding development strategies have become the key to promoting village development and optimizing village layout. This study established a comprehensive framework containing natural and socioeconomic aspects to identify the villages' development types. The villages in Laiyang Couty were used to examine the framework. The study found that this framework can group the villages in more feasible and reliable ways.

A village is a complex of natural environment and socioeconomic system. Researchers have studied the identification of villages' development types by constructing the evaluation system of a village development classification index [57]. The natural environment is the background condition of rural development, and the evaluation of natural suitability is one of the preliminary basic works of rural development. Priority should be given to evaluate whether it is restriction for village development. If it is not restricted, the suitability and its degree will be evaluated. The natural suitability/restriction evaluation, undertaken by using the combined method of the modified multiplication-weighted summation model, made the evaluation results more objective and has feasibility for replication. Based on the "bucket" principle, the restriction of some factors played decisive roles in the villages' natural suitability, and it was shown that some factors over their limit values were not suitable for the development of the village [44]. In terms of research methods,

an additive model is widely used, but it requires the evaluation factors to be independent of each other; otherwise, it will cause the information repetition among the evaluation factors, and the suitability degree cannot be objective and fully expressed. From the two perspectives of suitability/restriction and development level, the weighted summation model is supplemented with the multiplication model to make the evaluation result more scientific.

Villages with high natural suitability are usually accompanied by high levels of socioeconomic development. In addition to field survey and statistical data, the nighttime light data were added to more quantitatively evaluate the socioeconomic development of the study area. The results showed that the Luojia1-01 nighttime light image is suitable for the analysis of rural economic development, which expands the data source of this research and fills the gap in the application of the data in rural socioeconomic indicators. Due to the single use of nighttime light remote sensing data for the analysis of rural economy, the overall accuracy may not be high [38]. On the basis of the traditional quantitative economic statistical evaluation indexes, combined with nighttime light images to reflect the development of rural economy, this study realized the application of multi-source data fusion in rural socioeconomic indexes evaluation and also realized the test of nighttime light images in rural areas.

Although the framework used in this study can classify the village according to their natural and socioeconomic conditions, there are some limitations that shall be noted. In the application of Luojia1-01 nighttime light images, only one phase of image was selected; multiple nighttime light images can be fused for in-depth analysis in the future. In addition, when we selecting indicators, we mainly considered the key indicators affecting the development of rural areas and other relevant factors, such as special policies; the views of the villagers are considered less. The views of local residents also have an important impact on the future development of the villages, but the subjectivity is relatively strong, and it is not easy to quantify. How to set up a quantitative estimation system to accurately reflect the wishes of the local residents is left to be explored in the future study.

## 6. Conclusions

This study constructed a comprehensive natural–socioeconomic framework to identify villages' development types based on multi-source data; introduced modified multiplication-weighted summation and the coupling coordination degree model, and divided the villages into five development types in the study area. The proposed framework can provide a basis for rural revitalization and development planning. The main conclusions are listed below:

(1) The spatial distribution trend of natural suitability is "high in the southwest and low in the northeast," and the socioeconomic development level overall presents a "high–medium–low" distribution circle trend with the urban area as the center.

(2) The coupling coordination degree between natural suitability/restriction and socioeconomic development level is relatively high. The coupling coordination degree has obvious spatial distribution differences, with the urban area being the center, and the coupling being high in the south and low in the north. It gradually decreases from the inside to the outside.

(3) Villages' development types include urbanization development, construction development, agglomeration and linkage development, decreased or improved development, and relocation and integration development. A development suggestion of each type was proposed according to their advantages and disadvantages.

**Author Contributions:** Conceptualization, A.W. and X.Y.; data analysis, Y.L. and J.L.; investigation, Y.L. and C.G.; methodology, Y.L.; funding acquisition, T.Z. and A.W.; writing—original draft, Y.L.; review and editing, A.W. and X.Y. All authors have read and agreed to the published version of the manuscript.

**Funding:** This research was funded by the National Social Science Fund of China, grant number (20&ZD090) and Shandong Natural Science Foundation, grant number (ZR2019MD014).

**Institutional Review Board Statement:** Not applicable.

**Informed Consent Statement:** Not applicable.

**Data Availability Statement:** The data used to support the findings of this study are available from the corresponding author upon request.

**Acknowledgments:** The authors would like to thank the editors and reviewers for helpful comments.

**Conflicts of Interest:** The authors declare no conflict of interest.

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
