# Peer review of "Identification of Villages’ Development Types Using a Comprehensive Natural–Socioeconomic Framework"

_sustainability, doi:10.3390/su13137294_

Round 1

Reviewer 1 Report

The paper is interesting and the subject is relevant.

The findings may be very useful to local government and policy makers, as well as international readers.

Only minor remarks:

Please consider proofreading by the Authors themselves to eliminate minor language issues and possibly redundant sentences.

Line 68 - please consider providing a short description of what is Luojia1-01 image when mentioned in the paper for the first time

Lines 188-201 please consider explaining the ratings <2, 3 and 4

Lines 214 and 2019 please explain all the indexes in detail

Table 2 - Please consider explaining:

Why 2 points are assigned if there is no community health services and 1 if there is one?

Please consider explaining the negative and positive indexes better.

Table 4

Please consider explaining in detail why particular types of development of villages were chosen. What was the methodology behind this typology?

Author Response

Point 1: Line 68 - When you first make an error in your paper, briefly describe what the Lowe 1-01 picture is

Response 1: Thanks for your comments. We added more descriptions in the second paragraph of Section 2, as shown below.

The rural economic indicators involved in the evaluation index system are mainly based on agricultural economic indicators, such as the proportion of economic income of the elderly [28], total output value [29], total output value [30], proportion of non-agricultural output value [31], and level of village collective economic development [32]. Since the 1990s, nocturnal image data have been gradually used to reproduce the horizontal level of a region [33]. Social development, especially Luojia 1-01 image, can reflect social and economic development with finer spatial resolution, and is one of the world's professional noctilucent remote sensing satellites [34], which is used to detect the development of counties and other built-up areas. Hubei Province, Hunan and Jiangxi Province [35]. However, the application of Luojia 1-01 is concentrated in urban areas [36], involving spatialization of population [37], population distribution [38], index prediction [39], etc. Urgent. Therefore, it is imperative to explore the application of Luojia 1-01 night disaster scenarios in rural social and economic development.

Point 2: Lines 188-201 please consider explaining the ratings <2, 3 and 4

Response 2: Thank you. The explanations were provided as follows. Slope is an important factor affecting the formation and development of villages. The smaller the slope, the more convenient production and life, and the more conducive to the formation and development of the village. From this aspect, the slope has a negative impact on village's development, that is, the smaller the slope is, and the higher the score is. Therefore, referring to the national grading standard of cultivated land slope, it is determined that when the slope is less than 2 °, the highest value of this index was 4. When the slope is 2 °- 6°, the index was assigned 3. When the slope is 6 °- 15 °, the index was assigned 2. When the slope is 15 °-25 °, the index was assigned to 1. When the slope exceeds 25 °, it is not suitable for village development, and the index was assigned to 0.

Point 3: Lines 214 and 2019 please explain all the indexes in detail

Response 3: Thank you. All the indexes in section 4.3.3 were explained in detail, which was modified as follows:

4.3.3 Construction of modified multiplication-weighted summation evaluation model

This study took the village as the unit, and constructed modified multiplication-weighted summation evaluation model to evaluate suitability/restriction. It includes the following two parts:

Firstly, we used the series multiplication model [46] to evaluate restriction. As long as the index value of one factor exceeded the limit value, a village was classified as an un-suitable development zone and a restricted construction zone. The specific formula is shown in Equation (1), which was showed below:

 (1)

Where is the restriction index of a village x, m is the number of limiting value evaluation factors; k is the order of limiting value factor; is the score of the kth limit value factor of a village x.

For the other villages, the natural suitability index was calculated by the weighted summation model to evaluate their natural suitability. The calculation formula is shown in Equation (2), as follows:

  (2)

Where  is the suitability index of a village y, j is the number of suitability evaluation factors; i is the order of suitability factor;  is the score of the ith suitability factor of a village y; is the weight of the ith suitability factor.

Point 4: Table 2 - Please consider explaining:

Why 2 points are assigned if there is no community health services and 1 if there is one?

Response 4: Thank you very much for your valuable and constructive opinions. The availability of community health services is one of the important factors to the convenience of medical treatment. In order to unify the indicators and explain the results easily, this paper adopts negative indicators and negative assignment; 2 points are assigned if there is no community health service and 1 if there is one, and calculates the score through extreme value standardization. The experts remind us that we should think deeply and carefully, and the positive scoring is more direct. After modification, 0 points are assigned if there is no community health service and 1 if there is one. The evaluation results of negative assignment extremum standardization and positive assignment are consistent, but expert opinions make the article more logical. In addition to the indicators of community health service institutions, other similar negative indicators were adjusted to positive, and the positive score was given again, and the evaluation results were consistent with the original negative score. It mainly involves sections 4.2.2(Table 2).

Point 5: Please consider explaining the negative and positive indexes better.

Response 5: Thank you. The explanations were provided as follows. Positive index means that the higher the index value is, the better the suitability level or development level of the village is. On the contrary, the higher the index value is, the worse the suitability or development level of the village is. No matter whether the assignment is positive or negative, the assignment is consistent with the positive and negative attributes of the index; the index is quantified dimensionless through standardization.

Point 6: Table 4 Please consider explaining in detail why particular types of development of villages were chosen. What was the methodology behind this typology?

Response 6: Thank you. The explanations were provided as follows. Due to the different natural, socioeconomic conditions and the different development levels and modes of villages, there are some problems in the development of Chinese villages, such as unbalanced, inadequate and uncoordinated development. At present, China has put forward the strategy of rural revitalization, which requires local conditions and classified revitalization. Based on the national demand of implementing rural classified revitalization in China, this paper classifies the development of villages in the study area. Based on the natural suitability and the level of social and economic development, combined with the results of coupling coordination degree classification, aiming at different stages of development, facing the characteristics of villages, and referring to the previous research methods [57,58], this paper uses the combination matrix method of multi-dimensional characteristics to comprehensively determine the development type of villages.

Thank you very much for your comments.

Reviewer 2 Report

Brief  summary

This is an interesting article based on a research approach to the identification of villages’ development types using a specific framework that features natural-sociοeconomic aspects. This research features an evaluation of natural suitability which is applied (by a coupling model) in relation with the socioeconomic development status.  This study contributes positively as a guide for the government or local authorities as far as the official rural planning based on rural development and revitalization. Furthermore, its results could be used in future to detect the reasons why some areas were found unsuitable for village development and what needs to be changed regarding some of these conditions.

Broad  comments

Firstly, it is necessary to create a section that will be referred to as a literature review, above the Study Area part with knowledge from other international researches on similar issues. A lot of content in Introduction must be transferred to this new section and must be replaced with an introductory framework. Also the Data Sοurses paragraph would be more ideal to move to the Methodology. In the selection of evaluation indexes section I would suggest to include references of previous surveys that enlighten the statement of area’s status. The graphics in the figures have great clarity, a fact that provides to the reader a good sensation of the examined geography area. The factors / socioeconomic development indicators that were included and analyzed to determine the types of development of the villages are sufficient. The coupling coordination of indexes as far as the used model was compatible with the intended purpose. In the methodology section it is necessary to add some references that will support the mentioned assumptions and will strengthen the reasons for using this methodological approach.

Author Response

Point 1: First, it is necessary to create on top of the research area section a section that will be called a literature review, containing knowledge from other international studies on similar issues. Much of the introduction must be moved to this new section and replaced with an introductory framework.

Reply 1: Thanks for your comments. We have revised Section 1 and added a new literature review section, which reads"

  1. Introduction

The countryside is a natural and social complex formed over time, which is combined with the ecological environment and social and economic system [1]. Sustainable development depends on a range of socio-economic and natural environmental factors in rural areas. Different countries and regions have different policies for rural development due to their different levels of development. In order to improve the rural living environment, improve the living standard of rural residents, and narrow the gap between urban and rural development, many countries have adopted different methodologies to formulate sustainable development strategies [2] and plans.

In China, rural villages are experiencing prominent population loss and socioeconomic recession, especially those with harsh natural environments and remote locations [3]. Imbalanced development between urban and rural areas and inadequate rural development have become two major societal concerns. The Chinese government puts forward the strategy of rural revitalization and proposes that rural village development should be promoted according to the development status, location conditions and resource endowments of different villages in accordance with the law and evolutionary trends of village development [4].

Therefore, from the comprehensive perspectives of the natural suitability and the level of socioeconomic development, it has become an important issue for China to implement the strategy of Rural Revitalization. Taking Laiyang County of Shandong Province as an example, based on field investigation, statistical data and multi-source remote sensing images, this paper combines two-dimensional natural suitability / restriction with four-dimensional socioeconomic development level, constructs a comprehensive natural-socioeconomic framework, identifies and analyzes the development types of villages, and puts forward constructive development suggestions, Thus, it enriches the theoretical framework and system of village classification development, and provides scientific basis for making regional village development decisions.

The structure of this paper is as follows. Section 2 provides a literature review and proposes research direction of this study. Section 3 presents the construction of the comprehensive natural-socioeconomic framework, which forms the theoretical support for this study. Section 4 introduces the study area, data sources, research methods, and processing. Section 5 presents an analysis of the results to the natural suitability/restriction evaluation, socioeconomic development level evaluation, coupling coordination between natural suitability and socioeconomic development level, villages’ development types identification and discusses the theoretical contributions and future prospects of this study, and section 7 presents the conclusions.

  1. Literature review

Recent studies on rural areas mainly focus on the. neo-endogenous rural development [5], social innovation in rural governance [6], critical drivers for sustainable development in rural areas [7], the potential using efficiency of rural development [8], Rural development and natural resource management [9], urban–rural development [10], rural sustainable development [11-13], rural revitalization [14,15], rural transformation [16], rural tourism [17-19], wisdom [20,21] and beautiful village construction [22]. The research area is distributed in different countries and regions, including Europe, Russia, and post-Soviet countries, Italy, Korea, America and China. Different characteristics of the natural environment and regional socioeconomic conditions lead to great differences in the development of villages. Therefore, the development types of villages should be identified, so as to detect the feasible development and revitalization paths to better guide the direction of village development. Researches have been conducted on the identification of villages’ development types. For instance, the conditions of village relocation and consolidation were identified by constructing the suitability evaluation indexes system from three aspects (landscape index, topographical conditions, land utilization situation) [23]. Be-sides, some studies focused on the evaluation of villages’ development levels [24], e.g. the village development evaluation indexes system was constructed from three aspects of production, life and ecological function; or from four dimensions of living resources, industrial base, regional transportation and cultural ecological environment, to detect the types of villages [25-27].

Most of the rural economic indicators involved in the evaluation indexes system is based on the conventional economic statistical indicators, such as per capita disposable income of residents [28], economic density, total agricultural output value [29], per capita total output value of agriculture [30], ratio of non-agricultural output value [31], and development level of village collective economy [32]. Since 1990s, the nighttime light remote sensing data were gradually used to objectively reflect the level of social and economic development of a region [33]. Especially, the Luojia1-01 image can reflect social and eco-nomic development in finer spatial resolution which is one of the most professional nighttime light remote sensing data [34], and was used to detect the development of constructed areas, e.g. county level region in Hubei, Hunan and Jiangxi [35]. However, the application of Luojia1-01 is temporally concentrated in urban areas [36], involving population spatialization [37], population distribution [38], economic index prediction [39], etc. The application of nighttime light remote sensing images in rural areas remains to be an emergent task. Therefore, it is indispensable to explore the application of Luojia1-01 nighttime light image in rural socioeconomic development evaluation.

Studies on rural development employed the suitability or development levels of a single aspect to identify the villages’ development types. However, the village is the coupling of natural environment and social economy, a clear and systematic framework combining natural and socioeconomic factors is indispensable to the identification of the villages’ development types. According to the coupling and coordinating relationship between the natural environment and social economy, we should build a comprehensive framework and divide the development types of rural villages. The coupling coordination degree model is a model originating from physics, which is used to evaluate the coupling coordination relationship between different systems. It is currently a reference for measurements by different systems or system coupling coordination between internal elements. However, the existing studies were merely limited to new urbanization, rural settlement layout [40-42], etc. which is difficult to be extended to the identification of villages’ development types. Hence, it is important to strengthen the direction of study that couples the suitability and development of village development types. And it is the uppermost priority of the current research to explore the implementation path suitable for China's rural development.  

This study thus strived to proposed a comprehensive natural-socioeconomic frame-work of villages’ development types with natural–socioeconomic factors and multi-source data. The study was conducted and examined in the villages of Laiyang County, eastern China. Comprehensive consideration of natural suitability and socioeconomic development, the modified multiplication-weighted summation method and the coupling coordination degree model were used to establish the comprehensive natural–socioeconomic framework, which was used to identify the villages’ development types scientifically and then put forward rural revitalization suggestions. The study was expected to provide a reference for promoting the rural revitalization and village planning.”

Point 2: Also, the Data Sources paragraph would be more ideal to move to the Methodology. In the selection of evaluation indexes section, I would suggest to include references of previous surveys that enlighten the statement of area’s status. The graphics in the figures have great clarity, a fact that provides to the reader a good sensation of the examined geography area. The factors / socioeconomic development indicators that were included and analyzed to determine the types of development of the villages are sufficient. The coupling coordination of indexes as far as the used model was compatible with the intended purpose.

Response 2: Thank you. The data sources paragraph was moved to the section of Methodology, and the references was added in the selection of evaluation indexes section. The specific modify mainly involved in section 4.

Point 3: In the methodology section it is necessary to add some references that will support the mentioned assumptions and will strengthen the reasons for using this methodological approach.

Response 3: Thank you. The corresponding references have been added in the methodology section, as follows.

4.3. Evaluation of natural suitability/restriction

4.3.1. Selection of evaluation indexes

Villages are generally developed and formed in regions with flat terrain, rich resources, good geology and ecological security. Within a certain range, the smaller the slope and elevation, the more convenient are production and life, and thus it is more conducive to the formation and development of the village. The forest land coverage rate reflects the ecological quality, and the higher the forest land coverage rate is, the better is the ecological environment quality. Natural resources such as water, arable land and minerals form the material basis of rural development. The richer the natural resources, the greater the potential for rural development is. Good geological conditions and low risks for disasters are conducive to long-term rural development. However, villages are not suited to areas with bad natural conditions, such as excessive slopes, excessive elevation, and proneness to geological disasters, being located in ecological protection areas, being too close to water and being prone to floods. Therefore, these villages’ development should be restricted.

Laiyang County is located on low-mountain and hilly land with a peak altitude of 375 meters. It is rich in arable land, water and mineral resources, but their distribution is uneven. Considering the natural suitability, restrictions and natural environment conditions, seven evaluation indexes, i.e. the slope, altitude, geological conditions, forest coverage, cultivated land resources, distance from river/reservoirs and mineral resources, were selected to evaluate the natural suitability/restriction of villages, and five indexes (important ecological areas, distance from water, altitude[43], slope and risk of geological disaster) [44]were selected to evaluate restrictions for village development (Table 1).

4.3.3 Construction of modified multiplication-weighted summation evaluation model

This study took the village as the unit, and constructed modified multiplication-weighted summation evaluation model to evaluate suitability/restriction. It includes the following two parts:

Firstly, we used the series multiplication model [46] to evaluate restriction. As long as the index value of one factor exceeded the limit value, a village was classified as an unsuitable development zone and a restricted construction zone. The specific formula is shown in Equation (1), which was showed below:

  (1)

Where is the restriction index of a village x, m is the number of limiting value evaluation factors; k is the order of limiting value factor; is the score of the kth limit value factor of a village x.

For the other villages, the natural suitability index was calculated by the weighted summation model [47] to evaluate their natural suitability. The calculation formula is shown in Equation (2), as follows:

 (2)

Where  is the suitability index of a village y, j is the number of suitability evaluation factors; i is the order of suitability factor;  is the score of the ith suitability factor of a village y; is the weight of the ith suitability factor.

4.4 Evaluation of socioeconomic development

4.4.1 Selection of evaluation indexes

The socioeconomic development level of villages is mainly reflected in traffic location, resource endowment, cluster scale and basic facilities. Good traffic location facilitates the flow of people and resources. The higher the grade of road, the easier it is to travel and the more conducive to the development of a village. The closer a village is to a town, the easier it is to be driven by its development. The endowment of social resources, such as historical cultural resources and specialty products, is formed by long-term development, which is not only a sign of the level of rural development, but also an advantage for future development. The scale of village agglomeration, including the population, income level and village size, is a sign of the level of rural development. The more the population, the higher the income, the brighter the nightlight, and the larger the village scale, the higher the level of village development. Under the current trend of rural population outflow, excessive population outflow affects the development of villages. Village infrastructure construction includes educational conditions, medical conditions, hardened roads, water supply, gas supply, etc., which are all reflections of rural development and the quality of production and life. The more perfect the infrastructure, the more comfortable people's lives will be.

The study area has high-speed railway and various levels of highways. The travel conditions and the radiation effect of cities and towns also differ among different villages with different road grades and distances to cities and towns [48]. Laiyang pear, Wu Long calligraphy and other cultural resources and specialty products renowned at home and abroad; part of the villages was historical and cultural village, dinosaur fossil village and other specialty villages. Village cluster scale, such as population [49], economy and land agglomeration [50], is different. The study area attaches great importance to rural construction and has relatively good infrastructure [51], but there are differences among villages. Hence, 14 indicators were selected from four dimensions, including traffic location, resource endowment, cluster scale and basic facilities, to construct the evaluation indexes system of rural socioeconomic development in the study area, as shown in Table 2.

Thanks again for your constructive comments!

Reviewer 3 Report

A well written publication with solid foundation in the existing literature and previous research. However, there are numerous works that combine natural, socio-economic and institutional aspects to analyse the processes and dynamics of development in rural territories, so it should be made clear what this work contributes to these approaches.

The authors also place their study on a huge effort of data collection from different sources. The methods and results are well presented, it would be useful, however, to clarify some aspects. The use of figures is appropriate.

The discussion is solid, but although the conclusions are based on the results collected and rounding up the objective of the paper, more detail and depth would be appreciated.

For a better understanding, please explain a bit more detailed these aspects:

- Development types of villages have been identified, but the concept of Development has not been defined. There are key factors for development, such as the education or age of the population, or institutional factors that undoubtedly condition the development potential of a territory and have not been considered in the analysis.

- The territory is only defined by natural and socio-economic aspects, while the institutional arrangements that exist or may exist, and which are key to development strategies, have not been considered.

- The AHP methodology has been used to determine the weight of the indexes. It should be clarified how many and who were the qualified informants, as this is a key element.

- With which method the grouping into Development levels (high-level development area (HD), 341 medium-level development area (MD) and low-level development area (LD)) was done?

 - Proposals are made that can hardly fit in with what is called Development, a concept that integrates efficiency, equity and sustainability: "In order to adapt to the law of social development, this kind of village can gradually reduce the scale of village land according to the population migration and the withdrawal of homestead in the future, and use the land economically and intensively. The infrastructure and people's working and living conditions should be appropriately improved. And at the same time government should guide rural residents to transfer their agricultural land to large agricultural households and agricultural enterprises, expand the scale of agricultural production and develop appropriate economies of scale".

- The failure to take into account the perception of the local population seems to me to be a major weakness of the work. At least, in the identification of future strategies, the priorisation made by the population could be considered, using AHP for example

Author Response

Responses to reviewer 3

Point 1: The type of village development has been determined, but the concept of development has not yet been defined. There are a number of key development factors, such as the education or age of the population, or institutional factors, which undoubtedly affect the development potential of a Territory but are not taken into account in the analysis.

Response 1: Thank you. The explanations were provided as follows. Village development includes improvement of economic situation, infrastructure, education level and social system. This paper focuses on the natural conditions and socioeconomic conditions of village development. The socioeconomic conditions mainly focus on the location, resources, infrastructure and other objective indicators which are easy to quantify to reduce subjectivity. The education, age or system of the population is not directly considered, but the education convenience and medical convenience can indirectly reflect the education and medical level. These important factors will be further studied in the future.

Point 2: The territory is only defined by natural and socio-economic aspects, while the institutional arrangements that exist or may exist, and which are key to development strategies, have not been considered.

Response 2: Thank you. The explanations were provided as follows. China's social governance is mainly based on the county level, where the natural and socioeconomic conditions are different, but the social management system is basically the same. Therefore, this study mainly considers the impact of natural social economy. Next, we will fully consider the combination of the existing social management system, formulate future development strategies, and promote the sustainable development of all kinds of villages.

Point 3: The AHP methodology has been used to determine the weight of the indexes. It should be clarified how many and who were the qualified informants, as this is a key element.

Response 3: Thank you. Analytic hierarchy process is used to determine the index weight. 15 experts, including professors and experts in land use, urban and rural planning, land planning, land assessment and resource remote sensing, as well as representatives of natural resources bureau, planning institute and other relevant units, were invited to score the indicators. Based on the scoring of all experts, the judgment matrix is constructed to make the weight quantitative on the basis of qualitative, and the result is more objective. It was added in the Section 4.3.2 as follows.

4.3.2. Quantification of evaluation indexes

According to the requirements of village development based on natural environment conditions, the restrictions and suitability factors were quantified by referring to relevant standards and literature [45]. First, the restrictive factor was quantified, and the score of the restrictive factor was set to 0 or 1. If a limiting factor reached an extreme value and restricted the village’s construction, it was assigned 0, and the area was directly designated as an unsuitable area. An area below the extremum value was set as 1, it then continued to participate in the suitability evaluation. According to the influences of suitability factors on village construction and rural development, considering the reality of Laiyang County, the suitability factors were graded and assigned, and the extreme value method was adopted to conduct standardization treatment. The relative importance of each suitability factor on village construction and rural development was compared, and the weight of each suitability index was determined by analytic hierarchy process (AHP). 15 experts, including professors and experts in land use, urban and rural planning land planning, land assessment and resource remote sensing, as well as representatives of natural resources bureau, planning institute and other relevant units, were invited to score the indicators. Based on the scoring of all experts, the judgment matrix is constructed to make the weight quantitative on the basis of qualitative, and the result is more objective. The indexes and weights for natural suitability evaluation are shown in Table 1.

Point 4: With which method the grouping into Development levels (high-level development area (HD), 341 medium-level development area (MD) and low-level development area (LD)) was done?

Response 4: Thank you for the comment. The grouping into Development levels were grouped into three grades used natural break point method: high-level development area (HD), medium-level development area (MD) and low-level development area (LD). It was added to the first paragraph of Section 5.2 as follows.

  The socioeconomic development index F of the villages in Laiyang County ranged from 0.12 to 0.52, and were grouped into three grades using natural break point method: high-level development area (HD), medium-level development area (MD) and low-level development area (LD).  It can be seen from the Figure 4 that the level of rural socio-economic development presents obvious spatial differences, and presented a high–-medium–-low circle-layer distribution structure centered on the urban area. Statistical results found that there were 158 villages with HD (0.29 < F ≤ 0.52), accounting for 20% of the evaluated villages. They are mainly distributed in Guliu, Chengxiang, Jiangtuan towns. These villages have high-level traffic conditions, large areas, proximity to the urban area, expressways, more convenient village transportation, and good infrastructure, clear re-source advantages and high overall development. 366 villages occupied MD (0.21 < F ≤ 0.29), accounting for 47% of the evaulated villages. They are mainly distributed in Bo Linzhuang, Heluo, Xuefang, Yangjun and other towns. These villages had relatively good locations, relatively convenient external travel, some infrastructure, relatively abundant characteristic resources, high forest coverage rates and relatively high development levels. Another 250 villages were LD (F ≤ 0.21), accounting for 32% of the evaluated villages. They are concentrated in the northwest of Tangezhuang; the northeast of Wandi; and the southeast of Dakuang and other towns. These villages are far away from the city, have large slopes, imperfect infrastructure, the advantage of average scale (population, area, etc.), incomplete public facilities and low levels of overall development.

Point 5: Proposals are made that can hardly fit in with what is called Development, a concept that integrates efficiency, equity and sustainability: "In order to adapt to the law of social development, this kind of village can gradually reduce the scale of village land according to the population migration and the withdrawal of homestead in the future, and use the land economically and intensively. The infrastructure and people's working and living conditions should be appropriately improved. And at the same time government should guide rural residents to transfer their agricultural land to large agricultural households and agricultural enterprises, expand the scale of agricultural production and develop appropriate economies of scale".

Response 5: Thank you. The explanations were provided as follows. Due to the different natural, social and economic conditions in different regions, there are some problems in the development of China's urban and rural areas, such as unbalanced, inadequate and uncoordinated development, population outflow and hollowing out in some villages, idle land, low utilization efficiency, and poor sustainability of village development. In order to achieve common prosperity and sustainable development, China implements the strategy of rural revitalization, and proposes different types of village development, such as urban-rural integration. Based on the law of village development and China's national strategy of implementing rural classified revitalization, this paper carries out village development classification, and puts forward targeted development suggestions. For example, for small-scale villages with poor natural conditions and infrastructure, the government may guide the villagers who have moved to cities to withdraw from residential land and develop it into park or cultivated land, so as to reduce the scale of village construction land, make land use more intensive and efficient, improve the village environment, and leave the villagers with equal and fair living conditions with other villages and cities, so as to promote social efficient, fair and sustainable development.

Point 6: The failure to take into account the perception of the local population seems to me to be a major weakness of the work. At least, in the identification of future strategies, the priorisation made by the population could be considered, using AHP for example-

Response 6 Thank you. This paper focuses on the natural background and socioeconomic conditions of village development. The socioeconomic conditions mainly focus on the location, resources, infrastructure and other objective indicators which are easy to quantify to reduce subjectivity. The views of local residents also have an important impact on the future development of the village, but the subjectivity is relatively strong, and it is not easy to quantify. This paper does not consider it at present. In the future research, we will fully consider the wishes of the villagers, so as to make the research results more practical. Related content was added to the section 5.4.as follows.

5.4. Discussions

Though the framework used in this study can classify the village according to their natural and socioeconomic condistions, there are some limitations that shall be noted. In the application of Luojia1-01 nighttime light images, only one phase of image was selected, multiple nighttime light images can be fused for in-depth analysis in the future. In addition, when we selecting indicators, we mainly considered the key indicators affecting the development of rural areas and other relevant factors, such as special policies; the views of the villagers are considered less. The views of local residents also have important impacts on the future development of villages, but the subjectivity is relatively strong. How to set up a quantitative estimation system to accurately reflect the wishes of the local residents is left to be explored in the future study.

Thank you very much again for your comments and suggestions.

Round 2

Reviewer 2 Report

All proposed changes were carried out to the extent practicable.The authors showed special attention to what had been pointed out and there was a satisfactory improvement as far as this intersting article.

Reviewer 3 Report

The authors have responded to the recommendations made and the paper can be published in its present form.